# The Role of Indocyanine Green Fluorescence in Rectal Cancer Robotic Surgery: A Narrative Review

**DOI:** 10.3390/cancers14102411

**Published:** 2022-05-13

**Authors:** Elena Belloni, Edoardo Maria Muttillo, Salomone Di Saverio, Marcello Gasparrini, Antonio Brescia, Giuseppe Nigri

**Affiliations:** 1Oncologic Colorectal Surgical Unit and Robotic Surgery Unit, Department of Medical and Surgical Sciences and Translational Medicine, Sapienza University of Rome, 00189 Rome, Italy; elena.belloni@uniroma1.it (E.B.); edoardomaria.muttillo@uniroma1.it (E.M.M.); marcellogasparrini1@gmail.com (M.G.); antonio.brescia@uniroma1.it (A.B.); 2Department of General Surgery, ASUR Marche, AV5, Hospital of San Benedetto del Tronto, 63074 San Benedetto del Tronto, Italy; salo75@inwind.it

**Keywords:** rectal cancer, robotic rectal cancer, fluorescence surgery, anastomotic leakage, robotic surgery

## Abstract

**Simple Summary:**

Surgery remains the only curative treatment for rectal cancer, despite the progress in oncological therapies. The widespread use of robotic surgery drastically changed the approach to rectal cancer, reducing the burden of the procedure on the patient’s quality of life and allowing a faster recovery. Indocyanine green fluorescence has shown promising results in reducing severe surgical complications, such as anastomotic leak, that can delay the beginning of further chemo-radio treatments. The purpose of this descriptive review is to analyze the impact of indocyanine green fluorescence when applied in robotic surgery on short-term surgical outcomes for rectal cancer, providing a picture of the current literature on the issue, highlighting the heterogeneity of protocols and focusing on possible future development.

**Abstract:**

Background: In rectal cancer surgery, anastomotic leakage (AL) remains the most feared complication, with a frequency of up to 30% in non-high-volume centers. The preservation of proper vascularization is a key factor for successful anastomosis. The use of fluorescence with indocyanine green (ICG) as an intraoperative method to verify optimal perfusion is becoming an interesting tool in rectal surgery. Today, robotic surgery, together with the use of the intraoperative evaluation of the perfusion with ICG, could be a real strategy to deal with AL, allowing for a more delicate and less traumatic surgical technique. This strategy may allow for an extremely accurate surgery, and for optimal control of the proper vascularization of the rectum. Methods: The purpose of this descriptive review is to analyze the impact of fluorescence and robotic surgery on short-term surgical outcomes for rectal cancer. Results: We performed a systematic literature search using the PubMed, Embase and Cochrane library databases. The primary endpoints were to evaluate the application of ICG fluorescence in robotic rectal surgery and the rate of anastomotic leakage when using these technological implementations. The secondary endpoints were to evaluate the dosage of ICG and the timing of application by different surgeons. Conclusions: ICG fluorescence is an inexpensive and quick method to assess bowel perfusion, providing immediate feedback to the surgeon, even if its role has not been proven. A quantitative system must be systematically introduced to minimize the subjectiveness of the visualized image.

## 1. Introduction

Rectal cancer (RC) is the cause of one-third of the total deaths caused by colorectal cancer (CRC). With 45,230 new diagnoses expected in the USA, RC ranks as the third most frequent malignancy in both sexes and the second highest for mortality [1,2]. Despite improvements in treatment and associated improved survival, mortality remains substantial, and the incidence in young patients (>50 years) is increasing [1].

Today, surgical treatment is the focus of an increasingly multidisciplinary and multi-specialist program, which is fundamental to obtain positive results in terms of survival and local recurrences. At the same time, neoadjuvant therapy and radiotherapy increase the risk of surgical postoperative complications and weaken the patient, making rectal resections even more challenging, even for expert surgeons [3].

To date, the surgical approach tends to be minimally invasive (laparoscopic/robotic). In addition to the rectal anterior resection (RAR), the transanal approach has also developed; however, it remains controversial, and its indications are limited to selected cases.

Anastomotic leakage (AL) remains the most feared complication, with a frequency of up to 30% in non-high-volume centers. A recent consensus defined the term of colorectal fistula and classified it into three grades of severity based on subsequent treatment [4].

Many studies have focused on the analysis of predictive factors for AL, highlighting that it is a multifactorial process. In the various series, the most highlighted factors were the level of the anastomosis, preparative radiotherapy, the patient’s ASA score, blood loss and transfusions, improper vascularization, nutritional status, insufficient mobilization of the proximal colic stump and the number of stapler firing [5]. 

Certainly, the preservation of proper vascularization is a key factor for successful anastomosis. For this reason, the use of fluorescence with indocyanine green (ICG) as an intraoperative method to verify optimal perfusion is becoming a useful tool in rectal surgery. Many studies have shown the efficacy and safety of this easily reproducible and low-cost method, showing good specificity in recognizing visceral ischemia and the ischemic demarcation zone. For this reason, several changes in the strategy of performing a colorectal anastomosis have been described, allowing surgeons to perform safer anastomosis [6,7]. Together with fluorescence, the use of robotic technology, which is now increasing in various surgical departments, seems to allow better surgical dissections with better vision and greater precision, especially for sphincter-preserving or nerve-sparing surgery [8]. In fact, the robotic technique seems to combine the advantages of open surgery, such as the great dexterity and accuracy of the surgical technique, with the advantages of minimally invasive surgery in terms of early rehabilitation, postoperative pain and infections. Various strategies have been tested over time to identify and prevent a possible AL, such as the hydropneumatics test, endoscopic control and methylene blue test, which are important methods that have failed to significantly impact AL. Today, robotic surgery, together with the use of intraoperative evaluation of the perfusion with indocyanine green, could be a real strategy to deal with AL, allowing for a more delicate and less traumatic surgical technique. This strategy aims more to a sparing surgery, and an optimal control of the proper vascularization of the rectum. Certainly, to date, the main limitations are represented by the costs for robotic surgery and consequent need to select its use. Moreover, regarding the evaluation of fluorescence with ICG, the image analysis remains subjective and based on the surgeon’s experience. The purpose of this descriptive review is to analyze the impact of new technologies in oncological rectal surgery, specifically fluorescence and robotic surgery, on short-term surgical outcomes and to analyze the current protocol for the application of ICG.

## 2. Materials and Methods

### 2.1. Study Selection

We performed a systematic literature search in November 2021 using the PubMed, Embase and Cochrane library databases. The search algorithm included the following keywords: “Robotic rectal surgery AND Indocyanine OR Firefly OR ICG”. We followed the PRISMA statement criteria [9]. The research was restricted to articles in English and including exclusively human patients. All abstracts, studies and citations scanned were reviewed.

### 2.2. Data Extraction

Two independent investigators (EB, EMM) performed the search and examined potentially relevant articles. The following data were extracted: first author, year of publication, study design, number of patients and characteristics, both demographical and oncological, surgical technique, ICG dosage and dilution, timing of injection during surgery and in comparison with the colic resection and intraoperative and postoperative outcomes. 

### 2.3. Inclusion Criteria

The articles included in the review had to meet the following inclusion criteria: (1) the article must be written in English; (2) the full text must be available; (3) it must contain a previously unreported patient group (when analyzing patients from the same institution, the most informative and recent article was included); (4) it must present a retrospective or prospective study design; (5) it must include patients undergoing robotic rectal surgery; (6) it must report the use of ICG, specifying the dosage and timing; (7) it must present a complete report of the short-term outcomes, including mortality, morbidity and anastomotic leak. The Cochrane "Risk of Bias Assessment Tool", 6th edition, was used to calculate the risk of bias [10].

### 2.4. Exclusion Criteria

Articles were excluded from the review when: (1) they featured a series with less than 5 patients, (2) they consisted of case reports, (3) it was impossible to calculate the outcomes of patients undergoing robotic resection only when the series included also patients undergoing laparoscopic/open resections, (4) it was impossible to extract data about patients with sigmoid/rectal resections only or (5) the study included patients undergoing resection for benign diseases.

### 2.5. Outcomes of Interest

From all studies, the following relevant data were extracted: patient baseline characteristics (age, gender, BMI, type of procedure, ASA classification [11] preoperative treatment); intraoperative data (operative time, ICG dosage, timing of injection, changing in transection segment selection, ileostomy); postoperative outcomes (rate and type of complications, postoperative anastomotic leak, mortality, length of stay). 

The primary endpoints were to evaluate the application of ICG fluorescence in robotic rectal surgery and the rate of anastomotic leakage when using these technological implementations. The secondary endpoints were to evaluate the dosage of ICG and timing of application by different surgeons. 

### 2.6. Statistical Analysis

Data were tabulated using Microsoft Excel (Microsoft 365, version16.43), and a cumulative analysis was performed when possible. Categorical variables were extracted as numbers and reported as proportions. Anastomotic leakage was defined as “A defect of the intestinal wall at the anastomotic site leading to a communication between the intra- and extraluminal compartments. A pelvic abscess close to the anastomosis, even without any communication with the colonic lumen, should be considered as a leak” [4]. 

## 3. Results

### 3.1. Included Studies

The PRISMA flow diagram for systematic review is presented (Figure 1). Four studies [12,13,14,15], published between 2013 and 2020, were considered eligible. There was 100% agreement on data extraction between the two reviewers. One study was conducted in the USA [12], two in South Korea [14,15] and one in India [13]. Two studies were prospective [13,14], and two were retrospective [12,15]. No randomized trials were identified. A total of 1350 patients were included, of which 939 (69.5%) underwent robotic resections with the aid of ICG, and 411 (30.4%) without. The characteristics of the studies are summarized in Table 1. The quality of the studies was assessed using Newcastle-Ottawa Scale [16] scores (Appendix A).

### 3.2. Patient Characteristics

Of the 1067 patients, 652 (61%) were male, and the mean age was 55 years old. The mean BMI was 26. In total, 360 (33.7%) patients underwent preoperative CT-RT, 1056 (98.96%) patients underwent anterior rectal resections and 11 (1.03%) left hemicolectomy. All patients were affected by colorectal adenocarcinoma. 

### 3.3. Outcomes and ICG Application

Data are summarized in Table 2. The mean operative time varied from 232 to 490 min. The dosage of ICG ranged from 5 mg to 25 mg administrated as a fixed dose, while one study reported a pro/kg dosage [13]. No adverse reactions to ICG were reported. In two studies, ICG use was reported before and after the colic transection, while the other groups reported the injection of ICG only before the resection [12,13]. In 47 patients (6.8%), the use of ICG resulted in a change in the colic resection site previously selected. The percentage of ileostomy was reported only in two studies [12,14].

The 30-day mortality rate was 0%. Anastomotic leakage was reported in 16 patients (2.33%) whose procedures were conducted using ICG to evaluate tissue perfusion, and in 24 patients (5.8%) who underwent surgery without the aid of fluorescence. Hospitalization ranged from 4 to 13 days. 

## 4. Discussion

In the last decade, researchers on genetic and molecular aspects of cancer have largely modified oncological therapy, making it more targeted and effective. Rectal cancer treatment, however, relies currently on conventional methods such as chemotherapy and radiotherapy almost exclusively. Particularly, neo-adjuvant radiotherapy reduces the cancer burden and prevents local recurrence [17]. Timing between the end of the radiotherapy and the surgical intervention is crucial to obtain the best possible oncological results. A 4-week wait after the end of the radiotherapy provides the tumor immune response enough time to react completely and lowers the systemic toxicity of the radiation, resulting in higher regression rates and lower recurrence, compared to shorter intervals [18]. 

Nevertheless, surgery remains the only curative treatment of rectal cancer and the burden of the colorectal procedure has completely changed since the introduction and widespread use of mini-invasive surgery. These procedures have increasingly become more tolerable and have a faster recovery. However, the occurrence of a severe surgical complication, such as an anastomotic leak, can represent a critical step backward in the patient’s healing process. The resolution of this severe complication may necessitate a second surgical intervention, a derivative stoma and a delay in further oncological treatments, resulting in a worse oncological outcome [19].

Of the whole range of the colorectal procedures, rectal sphincter-saving surgery with low anastomosis has the highest rate of anastomotic leakage, with an incidence reported in the literature of up to 20% and the highest mortality rate [20].

Several patient-related elements have been identified that negatively affect the healing of the anastomosis, such as male gender, preoperative chemo-radio therapy, smoking, low anastomosis level, weight loss and tumor size [21]. Other factors are more directly influenced by the surgeon’s experience, such as the tension of the anastomosis, duration of the procedure and number of firings of the linear stapler [19].

Surgical teams perform different maneuvers to test the integrity of the anastomosis, such as intraoperative endoscopy and the air leak test. Moreover, adequate blood perfusion is considered one of the most important factors in the prevention of AL. A visual inspection of the intestinal stumps is sometimes unreliable [22] and highly subjective. Therefore, in the last decade, intraoperative angiography with ICG has become increasingly more important to assess microperfusions in several surgical fields. 

The number of studies published evaluating the use of fluorescence in colorectal surgery has drastically increased in the latest years, especially since the widespread use of robotic surgery and advanced laparoscopic systems that have integrated fluorescence detectors to exploit the near-infrared technology without requiring extra equipment in the operating room and without consuming additional time. 

Moreover, the routine use of ICG in colorectal procedures proved to be safe, with a really low rate of side effects and a cost-effective measure when considering the reduction in AL [23].

In the literature, many studies include both colic and rectal resections conducted indifferently in open, laparoscopic and robotic surgery. Therefore, the comparation on specific subsets of data results is particularly challenging [24,25,26,27]. For this reason, our review aimed to focus only on robotic cases undergoing a low anterior resection for cancer in order to provide data on a specific subgroup in which AL is a more frequent complication and has a greater impact on survival. 

From our data, the rate of AL was lower (2.2%) in patients undergoing resections with the aid of ICG compared to those who underwent a standard surgical procedure (5.8%). Both rates seem to be generally lower when compared to a recent meta-analysis on robotic and laparoscopic rectal resections, where AL was reported in about 10% of patients and no statistical significative difference was seen in AL rates between laparoscopic and robotic surgeries [28,29,30]. It is clear that no conclusion can be drawn from these data due to the several limitations that this narrative review presents, mainly represented by the lack of a control group, the small numerosity and the heterogeneity of patients. Moreover, the studies analyzed provide few data regarding tumor characteristics and surgical technique, with special regard to the level of IMA ligation or the complete mobilization of the splenic flexure. These factors are particularly critical in rectal surgery because they might have an impact on the onset of surgical complications. Bae et al. [14] focused specifically on the level of inferior mesenteric artery (IMA) ligation in low and ultralow anterior resections, presenting encouraging data on the positive impact of low arterial ligation on postoperative complications, especially AL, without compromising the lymphnodal harvest. However, this item could not be further supported because it is not clearly stated in the other studies included in the narrative review. 

Despite the several limitations of the data available, the implementation of the surgical rectal procedures with the aid of the analyzed technological innovations seems to be encouraging, and the results are in line with the recent literature. The data are also supported by a recent meta-analysis in which a reduction in AL is registered when ICG is applied to colorectal resections [31].

Yet, to our knowledge, only two randomized controlled trials [32,33] have evaluated the difference in AL between patients undergoing mini-invasive left colectomy/low anterior resection perfusion with the aid of near-infrared fluorescence imaging or with the standard surgical technique. No statistically significant difference in AL rates was observed between the two groups of patients in both studies. 

In almost 7% of patients considered in the review, the use of ICG resulted in a change in surgical strategy, meaning the resection line identified by the surgeon before the use of fluorescence was then reconsidered. In select cases, a second evaluation was performed after the transaction, resulting in a re-resection due to an unsatisfying image. Fluorescence was hence crucial in the perfusion assessment, avoiding potential leaks due to poorly perfused colic stumps.

One main concern about the application of ICG is the lack of consensus on the administrated dosage. As shown from our series, almost every author used a different protocol. To collect a large amount of data on the issue and establish a consensus, a European registry on fluorescence image-guided surgery (EURO-FIGS) [34] was created. As stated from a recent report, the mean dose used in colorectal cancer and inflammatory disease is 0.2 mg/kg. However, the majority of centers use a fixed dose of ICG, resulting in a lower concentration per kg in obese patients. Therefore, their recommendation is to use a calculated pro/kg dose, rather than a fixed value, to reach uniformity between different surgical centers and patients [7]. The International Society for Fluorescence Guided Surgery, on the other hand, recommends an intravenous injection of 3–3.5 mL diluted in 10 cc of saline solution to assess perfusion during colorectal resection [35]. A consensus on the matter is yet to be reached. 

In recent years, there has been a claim to develop a quantitative method to evaluate fluorescence. In most centers, the evaluation of fluorescence is only based on a subjective visual image perceived by the surgeon, therefore only considering the intensity of the signal. Consequently, it is difficult to compare data and improve protocols. Furthermore, fluorescence intensity is not a reliable parameter, since it is inversely correlated to the distance of the fluoroscope from the object and is influenced by ICG tissue diffusion. It has been proven that, over time, the fluorophore reaches ischemic areas, leading to an overestimation of the perfusion. All of these factors contribute to the need for a reliable, reproduceable dynamic evaluation of fluorescence. A recent technical implementation is fluorescence-based enhanced reality (FLER) technology, a software that analyzes the fluorescent signal over time and generates virtual perfusion cartograms, overcoming the possible limitations of fluorangiography [7,36,37].

Since it was first described in 1982 ref. [38], total mesorectal excision (TME) has become the standard of care in the management of rectal cancer because it minimizes the risk of local recurrence and eradicates potential mesorectal lymphatic involvement, which is frequent in cT2 or greater tumors [39,40]. The quality of TME represents a prognostic factor for cancer-related survival [41].

With the diffusion of mini-invasive surgical procedures, TME is now performed laparoscopically. Better results have been demonstrated in terms of perioperative outcomes, such as blood loss, postoperative pain and hospitalization. In addition, analogous findings in terms of local recurrence, circumferential margin positivity, the number of lymph nodes in the specimen and disease-free survival have been reported for both techniques. The long-term quality of life is also similar, independent of the surgical techniques [42,43].

One positive aspect of laparoscopic surgery is the magnification of the field and good visibility of the deep pelvis. Together with the reduction of bleeding, these factors help to obtain adequate nerve preservation [44].

However, laparoscopic rectal surgery is technically demanding, and it requires a large number of procedures performed to complete the learning curve. The surgical field is narrow; hence, the maneuverability of nonarticulated instruments is particularly limited. Moreover, there are several risk factors affecting the standardization of the technique, such as male sex, obesity and large tumors. These elements make the rectal transaction extremely challenging due to an even more restricted pelvic space available for tissue manipulation. For this reason, conversion to open surgery is high in rectal procedures, with a prevalence of 29% reported in the literature. Moreover, there is a risk for a positive circumferential resection margin and lower quality of TME, especially during the learning curve [28,45,46].

In rectal surgery, robotics aims to overcome some technical limitations of the laparoscopic approach thanks to better ergonomics and maneuverability of the articulated instruments, allowing for a better visualization of the surgical field and an easier exposure of the tissue plane and neurovascular elements. 

However, to date, RCT comparing robotic and laparoscopic techniques has found no statistically significant difference in the rates of conversion to open laparotomy in the positive circumferential resection margin, quality of TME, AL, number of harvested lymph nodes, perioperative morbidity and postoperative quality of life [29,47]. Between the reported postoperative advantages, there is a lower rate of pelvic autonomic nerve lesions—and, therefore, of postoperative sexual dysfunction—and reduced hospitalization compared with other surgical techniques [48]. Intraoperative blood loss is also reduced, as well as the conversion rate, while the operating time and cost of the procedure are consistently higher [30,45]. The budget limitation might be overcome with the achievement of surgical expertise that might decrease the operating time, and with the probable reduction in the costs of the materials. Moreover, when taking into consideration the reduction of postoperative complications, the shortened length of stay and the decreased conversion rate, up to a half of the laparoscopic one in some series, the exceeding expenses might be justified [49,50].

## 5. Conclusions

ICG fluorescence is an inexpensive and quick method to assess bowel perfusion, providing immediate feedback to the surgeon, even if its role has not yet been proven. A quantitative system must be systematically introduced to minimize the subjectiveness of the visualized image. 

The choice of surgical techniques should be tailored to the surgeon’s experience together with the patient’s characteristics, considering specific factors such as the level of the tumor, sex and BMI. 

In experienced hands, new technology in rectal cancer surgery seems to be promising in reducing hospitalization and postoperative complications such as anastomotic leaks, allowing a faster recovery for the patient and minimizing the impact on quality of life. 

## Figures and Tables

**Figure 1 cancers-14-02411-f001:**
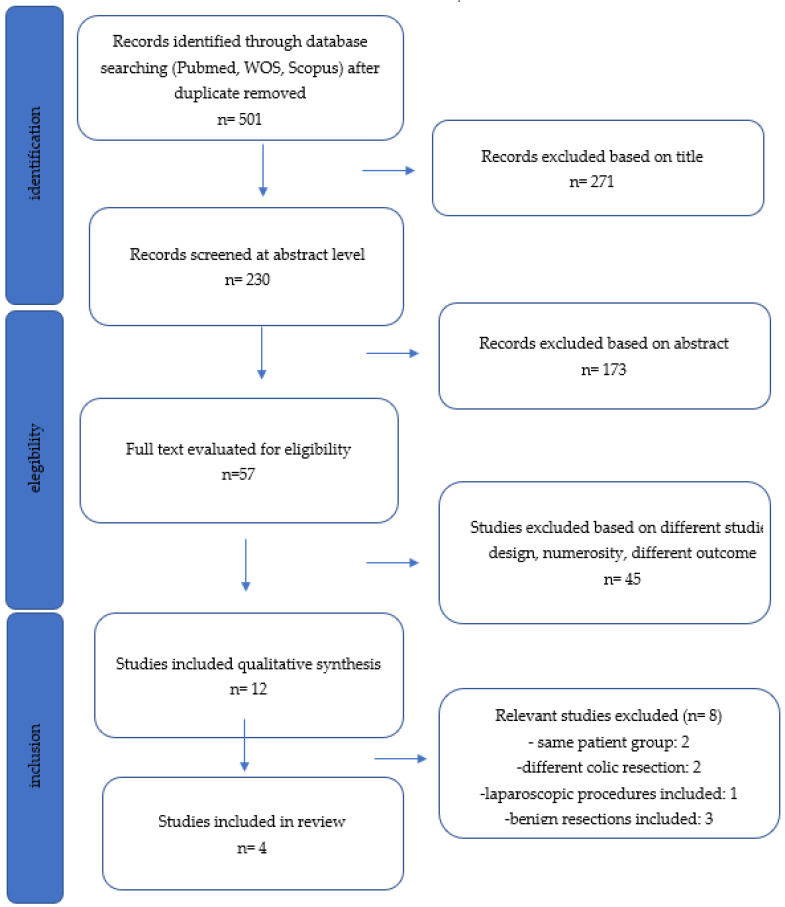
PRISMA flow diagram. The registry number is: reviewregistry1340.

**Table 1 cancers-14-02411-t001:** Studies selected and patients’ characteristics.

Author	Year	Country	ICG	No ICG	Age	Gender (M/F)	BMI	Neoadjuvant Therapy
*Jafari*	2013	USA	16	22	60.5	28/10	27	25
*Bae*	2015	South Korea	11	NR	42	6/5	22.9	7
*Kim*	2019	South Korea	609	359	57	586/382	23.5	287
*Somashekhar*	2020	India	50	NR	54.5	32/18	30 > 25	41

ICG: indocyanine green; M: male; F: female; BMI: body mass index.

**Table 2 cancers-14-02411-t002:** ICG application.

Author	ICG Dosage	Timing	Change of Plan	Anastomotic Leak ICG (%)	Anastomotic Leak No ICG (%)
*Jafari*	6–8 mg	Before	3 (19%)	1 (4.5%)	4 (18%)
*Bae*	5 mg	Before	NR	0 (0%)	NR
*Kim*	10 mg	Before and After	NR	15 (2.4%)	19 (5.3%)
*Somashekhar*	0.1–0.5 mg/mL/kg	Before and After	44 (88%)	1 (2%)	NR

ICG: indocyanine green; NR: not reported.

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
