# Peer review of "The Role of Indocyanine Green Fluorescence in Rectal Cancer Robotic Surgery: A Narrative Review"

_cancers, 2022, doi:10.3390/cancers14102411_

Round 1

Reviewer 1 Report

Interesting systematic review reinforcing the ICG valur at avoiding undesirable ALS. My only suggestion is to add p values at Table 2 and also to present the main finding, ie. the drop of AL incidence overall as an additional line at the bottom of the Table with a respective p value.

Author Response

To Cancers

Editorial Board

We would like to thank the reviewers for the comments and suggestions they reported. We appreciated them and modified the text accordingly.

Reviewer #1

We thank reviewer #1 one for his/her comments.

Giuseppe Nigri, MD, PhD, FACS, FRCS

Associate Professor of Surgery

Sapienza University of Rome, Italy

Reviewer 2 Report

Overall it is an interesting topic to investigate, however there are several major issues that need to be addressed.

Major issues:

  • The main endpoint of this narrative review is focused on anastomotic leakage (AL) reduction in rectal cancer surgery thanks to the use of ICG and robotic approach. While the relation between ICG and AL is clear and well discussed, it is absolutely not clear why robotic surgery should reduce AL. The rationale of reducing AL by robotic surgery is not explained but just stated in Introduction: the authors should better explain why they have focused on the use of ICG specifically in robotic-treated patients.
  • AL is affected by several features but the Authors considered only a few of them. Could the AL rate be extracted specifically in patients treated by neoadjuvant chemoradiation and in those treated by upfront surgery? Was a temporary stoma fashioned in all patients? What about the AL rate with or without ostomy? In the considered studies, AL was diagnosed due to symptoms or an imaging was routinely performed in all patients? Patients with a temporary stoma might be asymptomatic despite the presence of AL, being “false negative” cases. All the above-mentioned considerations are really relevant, and explain why the authors could not proceed to calculate a p Value for the difference between ICG and no ICG patients. They could not assess the risk of bias, the heterogeneity and the weight of each study as in a meta-analysis (and 4 studies only were included!), therefore the reported p Value makes no sense and should be deleted. This is, as correctly named by the authors, a narrative review.
  • It would be also assessed if in the included studies patients were treated by high or low ligation of the IMA, if splenic flexure was mobilized, etc. If these data are unavailable, then the Authors should clearly state this as a limitation in these studies, and properly discuss about the paucity of data in this topic.
  • An innovative clinical application of ICG that needs to be considered is to guide lymphadenectomy at the IMA origin in rectal cancer. I think that this important clinical aspect should be discussed somewhere within the Discussion.

Minor issues:

  • Cost saving issues should also be considered and discussed. Liu RQ et al (2022), investigated the cost impact of this intervention and reports that the routine use of ICG, reducing AL, leads to overall cost saving. However, the overall quality of evidence is low and there is a clear need for prospective, randomized controlled trials.

Author Response

To Cancers

Editorial Board

We would like to thank the reviewers for the comments and suggestions they reported. We appreciated them and modified the text accordingly.

Reviewer #2

  • Reviewer #2 wrote: “The main endpoint of this narrative review is focused on anastomotic leakage (AL) reduction in rectal cancer surgery due to the use of ICG and robotic approach. While the relation between ICG and AL is clear and well discussed, it is absolutely not clear why robotic surgery should reduce AL. The rationale of reducing AL by robotic surgery is not explained but just stated in Introduction: the authors should better explain why they have focused on the use of ICG specifically in robotic-treated patients”.

We appreciated the Reviewer’s comment since it gave us the possibility to specify the object of the manuscript. The focus of the article was to provide a “screenshot” of the impact of new technologies in rectal cancer surgery. Robotic surgery and the use of ICG are growing, especially in colorectal surgery. This narrative review was aimed to show these new technologies, without making comparison with other techniques.

We tried to provide data on a specific subset of patients on which literature is lacking, since the studies often include other types of colic resections and mini-invasive patients are analyzed en-bloc. (294-299)

The aim of this narrative review was not to demonstrate that robotic surgery reduces AL. As showed in the ROLLAR study, robotic surgery is not superior to laparoscopic surgery in terms of AL, conversion rate or significative reduction of post-operative complications. There seems to be a positive trend in literature, especially when analyzing specific subgroups of challenging patients (male, obese, low rectal resections), where robotic surgery provides benefits in the technical gesture. Moreover, some advantages were observed in terms of reduced pelvic autonomic nerve lesions and hospitalization, as stated on line 388-393.

According to the Reviewer’s comments, we specified in the text (line 385-388; 300-303) that robotic surgery does not reduce AL.

  • Reviewer #2 wrote: “AL is affected by several features but the Authors considered only a few of them. Could the AL rate be extracted specifically in patients treated by neoadjuvant chemoradiation and in those treated by upfront surgery? Was a temporary stoma fashioned in all patients? What about the AL rate with or without ostomy? In the considered studies, AL was diagnosed due to symptoms or an imaging was routinely performed in all patients? Patients with a temporary stoma might be asymptomatic despite the presence of AL, being “false negative” cases. All the above-mentioned considerations are really relevant, and explain why the authors could not proceed to calculate a p Value for the difference between ICG and no ICG patients. They could not assess the risk of bias, the heterogeneity and the weight of each study as in a meta-analysis (and 4 studies only were included!), therefore the reported p Value makes no sense and should be deleted. This is, as correctly named by the authors, a narrative review”.

We included in this study only the patients undergoing rectal resections for cancer, excluding 3 studies otherwise eligible. The reason of this was specifically to have a more homogenous set of patients. It was not possible to extract the data specifically for patients receiving neoadjuvant chemoradiation (33,7% of total). Only two studies (Jafari and Bae) reported data on ostomy (respectively 2/38 in Jafari and 6/11 in Bae) and no specific information was given if AL occurred in these patients.  According the reviewer’s comments and the inability to extract these specific data we deleted the p value reported and we specified in the text (line 304-306) that no statistical analisys was performed.  

  • Reviewer #2 wrote:” It would be also assessed if in the included studies patients were treated by high or low ligation of the IMA, if splenic flexure was mobilized, etc. If these data are unavailable, then the Authors should clearly state this as a limitation in these studies, and properly discuss about the paucity of data in this topic”.

We would like to thank you for this comment, which is a very important point. Unfortunately, surgical technique (with special regard to splenic flexure mobilization and IMA ligation level) is not stated in every included study. According to reviewer’s comments we added the data available on the IMA ligation extracted from the studies and clearly state this additional limitation due to the lack of sufficient data (307-319).

  • Reviewer #2 wrote: ”An innovative clinical application of ICG that needs to be considered is to guide lymphadenectomy at the IMA origin in rectal cancer. I think that this important clinical aspect should be discussed somewhere within the Discussion”.

We thank the reviewer for the observation, since the use of ICG in lymph nodes mapping is a very hot topic. However, the aim of our study was to analyze the role of ICG in rectal surgery to assess blood flow only, before and after performing the anastomosis. When guiding lymphadenectomy, ICG is usually injected endoscopically and often preoperatively. The presence of the dye at the level of the lesion or the spread of the dye itself could somehow modify the image of vascular perfusion. Moreover, the application of this innovative technique is more common in colic surgery, where it is used to guide CME. In rectal surgery, a high quality TME is generally sufficient to perform a complete excision of the lymphatic tissue.

Minor issues:

  • Cost saving issues should also be considered and discussed. Liu RQ et al (2022), investigated the cost impact of this intervention and reports that the routine use of ICG, reducing AL, leads to overall cost saving. However, the overall quality of evidence is low and there is a clear need for prospective, randomized controlled trials.

Thank you for your comment; we added this sentence and the relative citation to the text (line 291-292).

Giuseppe Nigri, MD, PhD, FACS, FRCS

Associate Professor of Surgery

Sapienza University of Rome, Italy

Round 2

Reviewer 2 Report

.